# ROYAL SOCIETY
# OPEN SCIENCE

# Research

geochemistry/petrology/volcanology

radon and thoron, zeolitized tuff, mineral dehydration, dilution by CO2 degassing

**Author for correspondence:**
Silvio Mollo
e-mail: silvio.mollo@uniroma1.it

# Carrier and dilution effects of CO$_2$ on thoron emissions from a zeolitized tuff exposed to subvolcanic temperatures

Silvio Mollo[1,2], Piergiorgio Moschini[1], Gianfranco Galli[2], Paola Tuccimei[3], Carlo Lucchetti[1], Gianluca Iezzi[4,2] and Piergiorgio Scarlato[2]

[1]Dipartimento di Scienze della Terra, Sapienza – Università di Roma, P.le Aldo Moro 5, 00185 Roma, Italy
[2]Istituto Nazionale di Geofisica e Vulcanologia, Via di Vigna Murata 605, 00143 Roma, Italy
[3]Dipartimento di Scienze, Università 'Roma Tre', Largo S. L. Murialdo 1, 00146 Roma, Italy
[4]Dipartimento di Ingegneria and Geologia, Università G. d'Annunzio, Via dei Vestini 30, 66013 Chieti, Italy

(ID) SM, 0000-0002-1448-0282

Radon ($^{222}$Rn) and thoron ($^{220}$Rn) are two isotopes belonging to the noble gas radon (*sensu lato*) that is frequently employed for the geochemical surveillance of active volcanoes. Temperature gradients operating at subvolcanic conditions may induce chemical and structural modifications in rock-forming minerals and their related $^{222}$Rn–$^{220}$Rn emissions. Additionally, CO$_2$ fluxes may also contribute enormously to the transport of radionuclides through the microcracks and pores of subvolcanic rocks. In view of these articulated phenomena, we have experimentally quantified the changes of $^{220}$Rn signal caused by dehydration of a zeolitized tuff exposed to variable CO$_2$ fluxes. Results indicate that, at low CO$_2$ fluxes, water molecules and hydroxyl groups adsorbed on the glassy surface of macro- and micropores are physically removed by an intermolecular proton transfer mechanism, leading to an increase of the $^{220}$Rn signal. By contrast, at high CO$_2$ fluxes, $^{220}$Rn emissions dramatically decrease because of the strong dilution capacity of CO$_2$ that overprints the advective effect of carrier fluids. We conclude that the sign and magnitude of radon (*sensu lato*) changes observed in volcanic settings depend on the flux rate of carrier fluids and the rival effects between advective transport and radionuclide dilution.

# 1. Introduction

Radon (*sensu lato*) is a noble, rare, inert gas produced by radioactive decay of nuclides that naturally occur in groundwaters, rocks and sediments of the Earth. This environmental radioactivity is determined by three isotopes decaying to daughter nuclides by emitting alpha particles: (i) $^{222}$Rn (radon), a product of $^{238}$U decay series ($^{226}$Ra is the direct parent nuclide) with half-life of 3.823 days; (ii) $^{220}$Rn (thoron), a product of $^{232}$Th decay series ($^{224}$Ra is the direct parent nuclide) with half-life of 56 s; and (iii) $^{219}$Rn (actinon), a product of $^{235}$U decay series ($^{223}$Ra is the direct parent nuclide) with half-life of 4 s. Because of its very short half-life and low activity, $^{219}$Rn is not employed in geochemical exploration. Conversely, $^{222}$Rn and $^{220}$Rn are widely monitored in volcano-tectonic settings, in order to discriminate areas with slow and fast gas transport or shallow and deep radiogenic sources [1–6]. A great virtue of $^{222}$Rn–$^{220}$Rn monitoring consists of their different half-lives, in conjunction with an identical geochemical affinity due to negligible isotopic fractionation between heavy radon and thoron isotopes, with mass difference of only 0.01%.

In the past decades, several experimental studies have investigated how $^{222}$Rn–$^{220}$Rn emissions measured from different lithologies change upon the effect of different laboratory-controlled variables, such as deformation regimes, temperature changes, mineralogical and chemical reactions [7–14]. In this respect, there is increasing knowledge that $^{222}$Rn–$^{220}$Rn anomalies are not univocally precursors of crust deformation and rupture, given that several radioactive phenomena may manifest a distinctive non-tectonic origin [15]. As a consequence, the marked increase of $^{222}$Rn–$^{220}$Rn signal in a monitored area is not imperatively symptomatic of impending earthquakes and volcanic eruptions [16–19].

The geological complexity of crustal lithologies may exert local filtering effects on the transport of radionuclides to the ground surface. These effects are exacerbated in volcanic areas, where the role played by carrier gases is combined with a number of thermally activated meteoric, hydrothermal, metasomatic and metamorphic reactions [20,21]. The final impact can be a decoupling between volcano-tectonic crises and $^{222}$Rn–$^{220}$Rn emissions, leading to transient, spatially heterogeneous, radioactive signals [4,5,22].

In a recent work, Mollo *et al.* [11] have experimentally documented the complex relationship between $^{220}$Rn emissions and zeolitized rocks, which typically originate by hydrothermal alteration in subvolcanic environments. Zeolites have the capability of adsorbing high amounts of water (up to approx. 25 wt%) in their structural cages and channels. Thermally activated dehydration reactions induced by shallow magmatic injections and dike intrusions may cause the release of water from zeolites, with important carrier effects on the transport of $^{220}$Rn radionuclides through the rocks [23,24]. The experiments presented in Mollo *et al.* [11] were designed to isolate the effect of water absorption–desorption phenomena on the background $^{220}$Rn signal measured under $CO_2$-free conditions.

However, $CO_2$ is an important carrier gas for radon (*sensu lato*) in volcanic settings [2,3,6], owing to its very low solubility in ascending magmas [25]. A close relationship between $^{222}$Rn–$^{220}$Rn and $CO_2$ emissions denotes rapid fluid transport along faults owing to an increasing concentration of radionuclides in the shallow soil/rock. Consequently, gas migration through lithologies with different petrophysical characteristics may also produce coupling/decoupling phenomena between $^{222}$Rn–$^{220}$Rn and $CO_2$ emissions [26]. There are volcanic conditions in which $CO_2$ fluxes through permeable (fractured/vesiculated) lithologies are high enough to overwhelm the radiogenic source, thus leading to a diluted $^{222}$Rn–$^{220}$Rn signal. For example, geochemical measurements carried out at the summit areas of Mt Etna volcano (Sicily, Italy) and along active faults on its flanks show progressive $^{222}$Rn–$^{220}$Rn dilution phenomena as the $CO_2$ flux increases [2,3].

In this study, we have re-designed the experimental set-up used by Mollo *et al.* [11] to conduct new real-time, long-term measurements on the transport of $^{220}$Rn nuclides emitted from the zeolitized rock under the carrier effect of variable $CO_2$ fluxes. We observe that $^{220}$Rn emissions are the expression of counterbalancing effects related to the physical (i.e. pore structure and gas advection) and chemical (i.e. adsorption–desorption of fluid molecular components) changes of the subvolcanic lithologies. Because of such complexities, the radon (*sensu lato*) signal does not necessarily increase in the presence of carrier gases of magmatic origin, providing explanation for spatio-temporal decoupling of radioactive patterns recorded along fault systems, diffuse degassing structures, fumaroles, and in the proximity of hydrothermal craters.

# 2. Methods

## 2.1. Experimental system and conditions

Thoron experiments were conducted at the HPHT Laboratory of Experimental Volcanology and Geophysics of the Istituto Nazionale di Geofisica e Vulcanologia (INGV) in Rome (Italy). A detailed

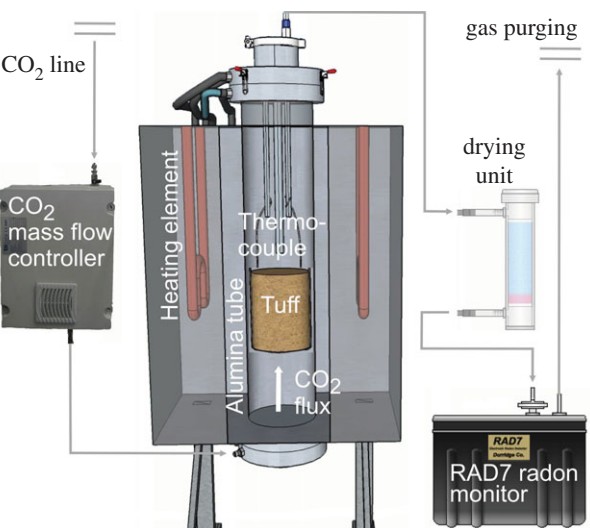

**Figure 1.** Sketch of the experimental system consisting of a vertical furnace equipped with a gas-impermeable alumina tube in which a cylindrical tuff sample is suspended. Four heating elements and one factory-calibrated thermocouple control the temperature with ±3°C uncertainty. A digitally controlled $CO_2$ mass flow meter operates from 0.01 to 3 l min$^{-1}$ with ±1.5% accuracy. An iron steel tube is inserted into the lower flange and fluxes a pure 100% $CO_2$ gas in the vertical furnace. A Teflon tube is inserted into the upper flange and connects the vertical tube with both a gas-drying unit ($CaSO_4$ desiccant with 3% $CoCl_2$, as an indicator) and the RAD7 (Durridge Company Inc.) counting system.

description of the experimental system is reported in Mollo *et al.* [11]. Briefly, the radon apparatus consists of a vertical furnace equipped with a gas-impermeable alumina tube in which a cylindrical *Tufo Rosso a Scorie Nere* (TRSN) (see below for more details) tuff sample is suspended (figure 1). Four heating elements and one factory-calibrated thermocouple control the temperature with ±3°C uncertainty. For the purpose of our experiments, the vertical furnace has been implemented with a digitally controlled $CO_2$ mass flow meter operating from 0.01 to 3 l min$^{-1}$ with ±1.5% accuracy (figure 1). Gas-tight flanges seal the upper and lower portions of the alumina tube. A steel tube is inserted into the lower flange and fluxes a pure 100% $CO_2$ gas in the vertical furnace. A Teflon tube is inserted into the upper flange and connects the vertical tube with both a gas-drying unit ($CaSO_4$ desiccant with 3% $CoCl_2$, as an indicator) and the RAD7 (Durridge Company Inc.) counting system (figure 1). $CO_2$ transported to the RAD7 is purged via a gas pipeline pre-installed at the HPHT Laboratory (figure 1).

RAD7 is set in a pump on (0.8 l min$^{-1}$) configuration and is equipped with a solid-state detector for alpha counting of $^{220}$Rn progeny. The electrostatic detector collects the charged ions and discriminates the electrical pulses generated by their alpha decays. This allows us to only measure the short-lived $^{216}$Po to rapidly determine the $^{220}$Rn. The radioactive equilibrium between $^{216}$Po and $^{220}$Rn is achieved in a few seconds owing to the very short half-life of $^{216}$Po (0.15 s). A period of 30 min is selected as the acquisition time of a single measurement cycle. To minimize the uncertainty associated with $^{220}$Rn signal, 48 cycles per experiment (30 min acquisition time per cycle) were collected. For each dataset, the uncertainty of the mean at 95% confidence level was then calculated.

For the thoron experiments, a TRSN cylindrical sample (60 mm in diameter and 200 mm in length for a total weight of 1784 g) was fitted to the alumina tube, ensuring that $CO_2$ gas was prevalently fluxed across the porous structure of the material. TRSN was kept at 110°C for 72 h to remove moisture and, subsequently, the sample was heated to the final target temperature. After 48 h of heat homogenization, $^{220}$Rn activity concentration was monitored for very long-term measurements of 24 h relative to the very short half-life of $^{220}$Rn nuclides. Three target temperatures of 170°C (EXP1), 230°C (EXP2) and 450°C (EXP3) were adopted, in accord with the thermally induced devolatilization reactions of zeolite minerals which characterize the rock structure (see below). For each target temperature, the $CO_2$ flux was stepwise increased by 0.5 l min$^{-1}$, starting from 0 to 3 l min$^{-1}$, corresponding to the maximum operating condition of the mass flow meter. $^{220}$Rn reading was corrected for: (i) the percentage of $CO_2$ in the system, which reduces the electrostatic collection of thoron daughters [27]; (ii) the decay occurred during gas transport through the tubing circuit [11]; and (iii) the effect of absolute humidity on the efficiency of the silicon detector [28]. A detailed description of the correction procedure adopted for the experimental system can be found in Mollo *et al.* [11], while $^{220}$Rn activity concentrations and related uncertainties are reported in the electronic supplementary material.

## 2.2. Petrochemical analyses

Microchemical and textural analyses were carried out at the HPHT Laboratory of INGV using the back-scattered electron mode of a field-emission gun-scanning electron microscope Jeol 6500F equipped with an energy-dispersive spectrometer detector and a JEOLJXA8200 electron probe micro-analyser equipped with five wavelength-dispersive spectrometers (15 kV accelerating voltage and 10 nA beam current, following the analytical conditions and standards reported in Iezzi *et al.* [29]).

Total porosity was measured on cylindrical tuff specimens (4 cm in diameter) loaded in a helium pycnometer AccuPyc II 1340 (Micromeritics Company) with ±0.01% accuracy.

Thermogravimetric (TG) and differential thermogravimetric (DTG) analyses were performed with an SDT Q600 analyser (TA Instruments). The dual-beam design virtually eliminates beam growth and buoyancy contributions to the underlying signal. For the analyses, approximately 30 mg of powdered rock sample were heated in an air atmosphere (20 ml min$^{-1}$) using a Pt crucible at a rate of 10°C min$^{-1}$ up to 1000°C.

X-ray powder diffraction (XRPD) patterns were collected with a Siemens D5005 diffractometer operating in the $\theta$–2$\theta$ vertical configuration, equipped with a Ni-filtered CuK$\alpha$ radiation and installed at the Department of Ingegneria and Geologia of the University of G. d'Annunzio. Each XRPD spectra were recorded between 4° and 80° of 2$\theta$, with a step scan of 0.02° and a counting time of 8 s. Crystalline phases were identified by a search-match comparison with the commercial Inorganic Crystal Structure Database (ICSD). Lattice parameters were refined with the Le Bail method and further phase abundances (wt%) were derived by the Rietveld method, as reported in Mollo *et al.* [11].

# 3. Results and discussion

## 3.1. Characterization of the zeolitized tuff

TRSN belongs to the main body of a reddish ignimbrite from Vico volcanic apparatus (Latium, Italy). This pyroclastic tuff contains crystals, ashes and black pumices and has a total porosity of approximately 47%, owing to the concurrent presence of macro- (from 1 mm to 1 cm) and micro- (from 1 to 100 µm) vesicles. Macroporosity is dominated by distinctive lithophysae, which are cm-scale cavities formed by trapped pockets of gas within the cooling volcanic ash (figure 2*a*). In such lithophysae-rich tuff, the internal structure is almost entirely characterized by connected pores, thereby the material is highly permeable to gas flow [9]. At the micrometer scale, TRSN also shows a spongy-like texture resulting from post-emplacement zeolitization phenomena caused by hydrothermal alteration (figure 2*b*). Heat treatment of TRSN induces strong dehydration reactions of zeolite minerals [11] and development of devolatilization features in the rock matrix (figure 2*c*).

A homogeneous matrix of fine glass shards resulting from rapid quenching of vesicular magma fragments at the time of eruption and zeolite crystals of chabazite are the most abundant phases (approx. 78 wt%) of TRSN. The XRPD pattern is prevalently resolved by the crystalline structure of chabazite (figure 2*d*), a hydrated Na and Ca aluminosilicate framework mineral and one of the most porous natural zeolites. A DTG curve shows a faint peak at 95°C (figure 2*e*) that corresponds to hydration water ($H_2O^+$) resulting from atmospheric humidity adsorbed on the TRSN grains. The structural water ($H_2O^-$) of chabazite is released via thermally induced crystallochemical changes caused by lattice deformation [30–32] and is identified by three endothermic peaks at 170, 230 and 450°C (figure 2*e*). TG analysis quantifies both $H_2O^+$ and $H_2O^-$ contents as percentage mass loss (figure 2*f*). The amount of moisture is not negligible and corresponds to 3.6 wt% $H_2O^+$. For the purpose of our thoron experiments, any outgassing effect related to the adsorbed atmospheric humidity was eliminated by heat treatment at 110°C for 72 h. As a consequence, the release of water was entirely controlled by the thermal dehydration of chabazite as a function of the devolatilization of 5.5 (EXP1), 3.6 (EXP2), and 2.1 (EXP3) wt% $H_2O^-$ (figure 2*f*).

## 3.2. Temporal evolution of $^{220}$Rn: dehydration versus dilution phenomena

Results from our experiments are plotted in the $^{220}$Rn signal versus dwell time diagram (figure 3). For the sake of completeness, these new data are compared with $^{220}$Rn emissions (890–2650 Bq m$^{-3}$) from $CO_2$-free experiments of Mollo *et al.* [11] conducted on a fully dehydrated tuff sample exposed to the same thermal conditions (170–450°C). As a general rule, the onset of chabazite devolatilization causes the

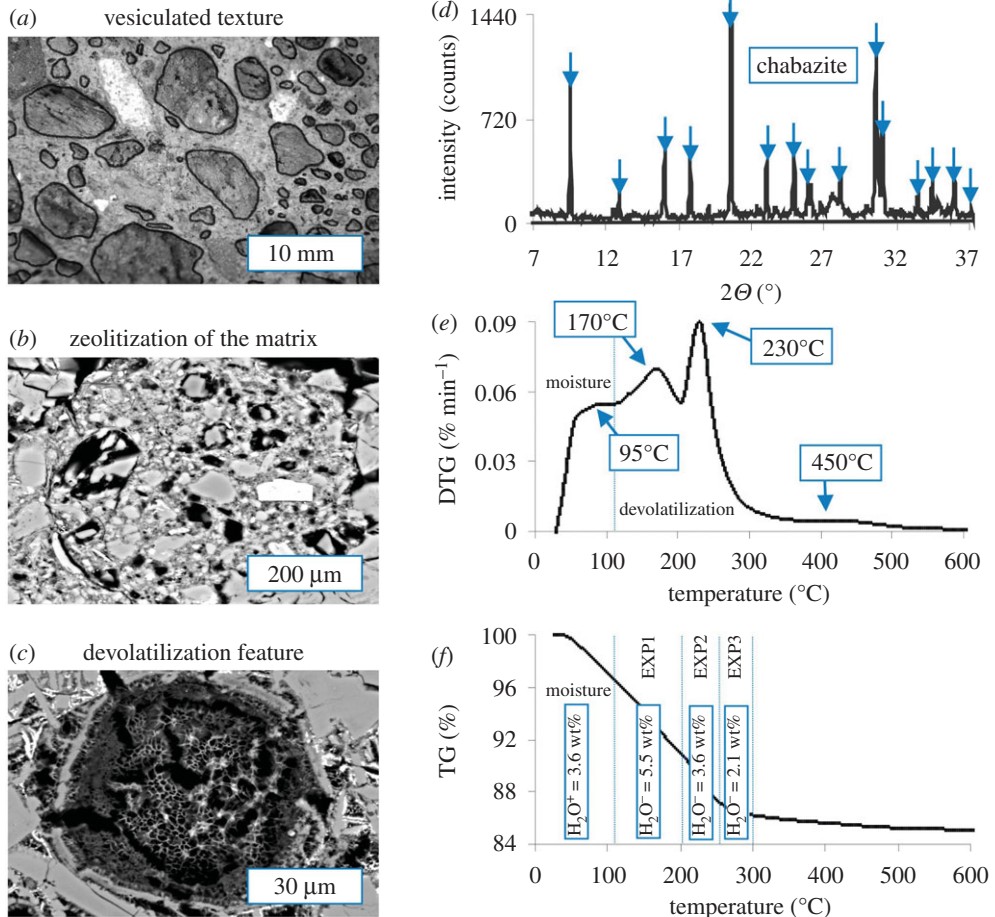

**Figure 2.** Texture of the tuff characterized by abundant vesicles (*a*), zeolitization of the matrix (*b*), and thermally induced devolatilization features (*c*). The tuff is mainly composed of zeolite and the most part of measured Bragg reflections are fitted with the ICSD chabazite standard (blue arrows) (*d*). Differential thermogravimetric peaks identify the onset temperatures of multiple dehydration events (*e*). A thermogravimetric curve tracks the amount of hydration water ($H_2O^+$) owing to moisture and structural water ($H_2O^-$) released by zeolite dehydration (*f*).

$^{220}$Rn signal to increase by one order of magnitude with respect to the signal observed under anhydrous conditions (figure 3). This points out that the simple enhancing effect of temperature on the diffusion of $^{220}$Rn nuclides through the material is greatly subordinate to the carrier effect of water vapour produced by chabazite dehydration (cf. [11]).

As the $CO_2$ flux increases from 0 to $1 \, l \, min^{-1}$, $^{220}$Rn emissions further increase monotonically up to a maximum value that is proportional to the amount of $H_2O^-$ released via chabazite dehydration (figure 3). More specifically, $^{220}$Rn emissions increase by approximately 165%, approximately 82% and approximately 25% for EXP1 (5.5 wt% $H_2O^-$ released at 170°C), EXP2 (3.6 wt% $H_2O^-$ released at 230°C) and EXP3 (2.1 wt% $H_2O^-$ released at 450°C), respectively. The inverse relationship between $^{220}$Rn and $H_2O^-$ depicted by the dashed arrow in figure 3 points out that the thoron activity concentration is mutually controlled by: (i) the dehydration process of chabazite as a function of the activation temperature, and (ii) the adsorption–desorption of water molecules and hydroxyl ($OH^-$) groups on the grain surface of TRSN as a function of $CO_2$ flux.

Because water molecules obstruct the direct passage of thoron through the medium, vapour adsorption on the surface layer of interconnected pores may limit the diffusivity of $^{220}$Rn nuclides [33,34]. A slow thoron mobility is exacerbated by condensation phenomena of water molecules in the pore spaces upon the effect of capillary forces [11,35,36]. Furthermore, hydroxyl coverage takes place in the highly porous structure of TRSN [37] and hydroxyl groups are easily bonded to Si cations of the silica network to form silanol (Si–OH) groups [38–40]. This mechanism is particularly effective for silicate materials thermally treated between 150 and 1200°C, as for the case of the hydrophilic surfaces of silicate glass shards forming the TRSN cineritic matrix.

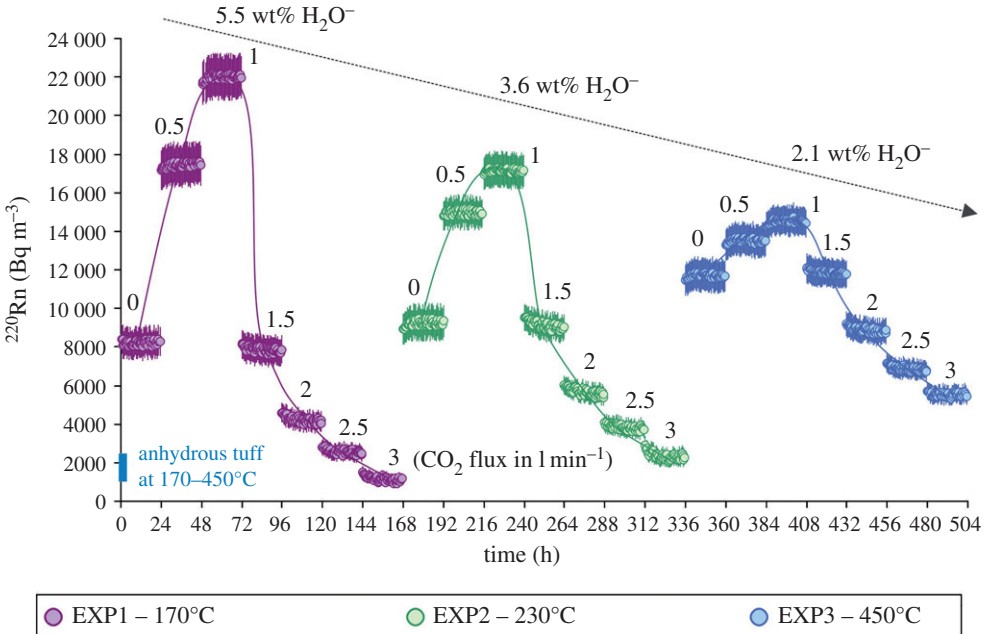

**Figure 3.** $^{220}$Rn versus time diagram showing the effect of temperature, water vapour release and $CO_2$ flux rate on the thoron signal. Data from this study are also compared with $^{220}$Rn emissions (890–2650 Bq m$^{-3}$) from $CO_2$-free experiments of Mollo et al. [11] conducted on a fully dehydrated tuff sample exposed to the same thermal conditions (170–450°C).

Water molecules and hydroxyl groups adsorbed on the glassy surface of macro- and micropores are physically removed under the increasing effect of $CO_2$ flux. Thus, most of the free-state $^{220}$Rn nuclides residing in the pore spaces are transported by the carrier fluid mixture through the medium and delivered to the detector. A relatively low $CO_2$ flux, in the range 0.5–1 l min$^{-1}$, acts as a physical and chemical agent for $^{220}$Rn mobility by permeating the pore structure of TRSN, desorbing water molecules and carrying $^{220}$Rn radionuclides through the medium.

The diagram of $^{220}$Rn signal versus $CO_2$ flux (figure 4) highlights one of the most important outcomes from our experiments, that is the systematic decrease of $^{220}$Rn emissions for $CO_2$ flux higher than the threshold value of 1 l min$^{-1}$. Such a declining trend is established irrespective of either the thermally induced dehydration of chabazite or the desorption of water molecules and hydroxyl groups. Rationally, the change in $^{220}$Rn behaviour is dictated by strong dilution phenomena that are more markedly observable when the $CO_2$ flux becomes so high that its magnitude overwhelms the advective effect of carrier fluids [2,3,41,42]. At $CO_2$ flux of 1.5 l min$^{-1}$, $^{220}$Rn emissions return to values almost comparable to those measured at 0 l min$^{-1}$ (figure 4), denoting an apparent competition between (i) $^{220}$Rn increase via chabazite dehydration, water desorption and advective transport, and (ii) $^{220}$Rn decrease by $CO_2$ flux in excess acting as a diluting agent for the concentration of radionuclides into the advective gas carrier.

When the $CO_2$ flux further increases to 3 l min$^{-1}$, $^{220}$Rn emissions dramatically decrease to minimum levels of approximately 1200 (EXP1), 2400 (EXP2), and 5600 (EXP3) Bq m$^{-3}$, confirming the strong dilution capacity of $CO_2$. However, the diluted $^{220}$Rn signal remains systematically higher than that (approx. 1000–2200 Bq m$^{-3}$) measured by Mollo et al. [11] on the anhydrous TRSN sample under $CO_2$-free conditions (figure 4). This finding points out that thermally induced devolatilization reactions of zeolites are still operative in the $CO_2$-saturated tuff and that the overall decrease of $^{220}$Rn emissions is the main expression of counterbalancing effects dictated by the physical (i.e. pore structure and gas advection) and chemical (i.e. adsorption–desorption of fluid molecular components) changes of the volcanic lithology.

## 3.3. Limitations and implications for natural volcanic settings

Radon–thoron laboratory experiments offer the advantage to isolate and analyse each single parameter controlling the radioactive signal, thus enabling the quantification of the sign and magnitude of $^{222}$Rn–$^{220}$Rn changes upon the influence of a specific effect [7–14]. It remains true that both physical and chemical phenomena observed at the laboratory scale represent an oversimplification of more

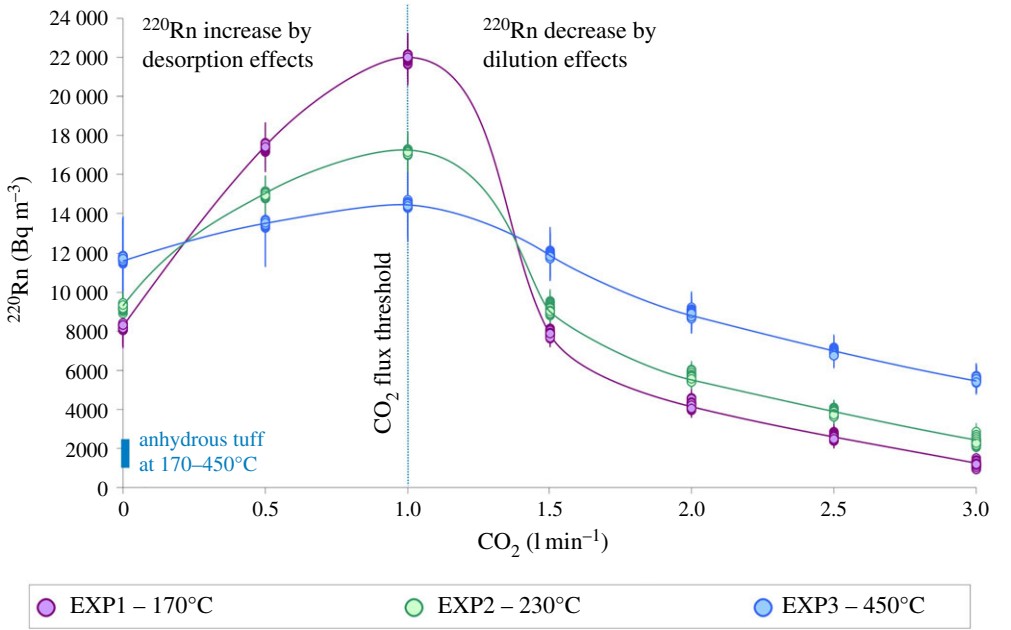

**Figure 4.** $^{220}$Rn versus $CO_2$ diagram showing two opposite trajectories owing to (i) thoron increase by desorption effects caused by expulsion of water molecules and hydroxyl groups adsorbed on the glassy surface of macro- and micropores and (ii) thoron decrease by dilution effects caused by $CO_2$ excess with respect to a hypothetical threshold value of $1\,l\,min^{-1}$.

complex geological conditions encountered in nature. Indeed, multiple processes are expected to influence the radioactive signal emitted from subvolcanic rocks before the gas reaches the ground surface, such as moisture, temperature, degassing, mixing, contamination, rock alteration, chemical reactions, soil radiogenic production and dilution effects (e.g. see Mollo *et al.* [25] for a comprehensive review). Under such circumstances, a direct comparison between experimental and natural $^{220}$Rn emissions is not a trivial task, given that geochemical data recorded in volcanic settings are mediated by the contributions of different magmatic and crustal sources and so any resultant radioactive signal cannot be univocally related to a single subvolcanic lithology.

In addition to the complexity of natural environments, we observe that there are only a few studies from the literature where $^{220}$Rn and $CO_2$ are concurrently investigated and, consequently, an overall comparison between experimental and natural signals is difficult to accomplish. A further complication arises when $CO_2$ fluxes used in the laboratory are converted from $l\,min^{-1}$ to $g\,m^{-2}\,d^{-1}$ taking into consideration the diameter (60 mm) of TRSN cylindrical sample, the molar mass of $CO_2$ ($44.01\,g\,mol^{-1}$) and the volume of one mole of gas ($22.41\,l\,mol^{-1}$). Results from calculations indicate that the putative natural $CO_2$ flux is up to two orders of magnitude higher than that generally associated with $^{220}$Rn emissions measured during soil gas survey. Moreover, the very short half-life (56 s) of $^{220}$Rn relative to that (3.823 days) of $^{222}$Rn must be taken into consideration for a better interpretation of the geochemical dataset. In volcanic settings, the gas carrier velocity through a subvolcanic succession rarely exceeds $100\,m\,d^{-1}$ [5] and, therefore, the travelling distances of $^{220}$Rn and $^{222}$Rn before complete decay of their radionuclides are 0.27 and 1900 m, respectively. This implies that, owing to its short half-life, $^{220}$Rn is useful only as a tracer of very shallow phenomena, such as porosity/permeability changes at depths of a few meters [2,3] and/or the presence of Th-rich mineralization close to the surface [25].

According to the above considerations and looking at several previous studies, we have tentatively interpolated $^{222}$Rn–$^{220}$Rn–$CO_2$ changes measured in natural volcanic settings, with the intention that the resulting dataset will be germane to other volcanic areas. Following this line of reasoning, we propose a general conceptual model that is independent of problems arising in scaling laboratory quantities to natural environmental conditions. The corollary of this approach relies on two important considerations: (i) $^{220}$Rn and $^{222}$Rn have the same geochemical behaviour, as long as heavy isotopes do not fractionate owing to their extremely low mass difference (0.01%); and (ii) $^{220}$Rn and $^{222}$Rn activity concentrations may substantially change under magma degassing conditions, accounting for their different half-life.

The selected geochemical dataset comprises measurements from the fumarolic activity of Sugás-Bai (Baraolt Mts, Eastern Carpathians, Romania; [43]), the Pernicana fault system (Mt Etna volcano; [3]) and

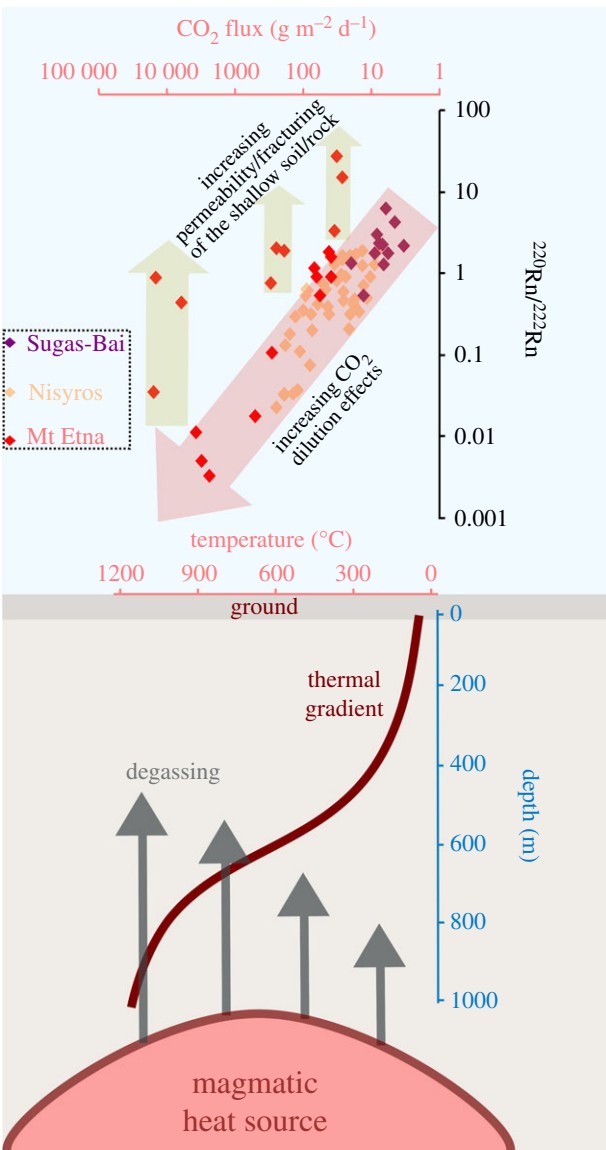

**Figure 5.** Comparison between $^{220}$Rn/$^{220}$Rn ratio and $CO_2$ flux measured in soil from three distinct environments: (i) in the area of fumaroles of Sugás-Bai (Baraolt Mts, Eastern Carpathians, Romania; [43]); (ii) along the Pernicana fault system, at the northeast flank of Mt Etna volcano (Sicily, Italy; [3]); and (iii) at the east side of a diffuse degassing structure and in the proximity of hydrothermal craters at Nisyros caldera (Aegean Arc, Greece; [6]). Subvolcanic thermal gradients and degassing phenomena from a putative magmatic reservoir located at depth of 1000 m below the ground surface were modelled by two-dimensional numerical simulations based on an explicit finite-difference scheme [44].

the degassing structures at Nisyros caldera (Aegean Arc, Greece; [6]). Some representative data are plotted in figure 5, also showing a schematic sketch of subvolcanic thermal gradients and degassing phenomena from a putative magmatic reservoir located at depth of 1000 m below the ground surface. Degassing occurs by pressure drop during the ascent of magmatic fluids through subvolcanic rocks and further fluid–soil interaction at the ground surface [45]. The source of heat and fluid transfer is a small volume (approx. 0.1 km$^3$) of magma that progressively cools over a temporal scale of 100 years (figure 5). Modelling was performed by two-dimensional numerical simulations based on an explicit finite-difference scheme [44]. Following the parameterization procedure reported in Mollo *et al.* [46], the input data used for the modelling was the initial temperature, bulk density, specific heat and thermal conductivity of magma (1150°C, 2700 kg m$^{-3}$, 1200 J kg$^{-1}$ K$^{-1}$, and 1.8 W m$^{-1}$ K$^{-1}$, respectively) and host rock (30°C, 2200 kg m$^{-3}$, 1150 J kg$^{-1}$ K$^{-1}$, and 3 W m$^{-1}$ K$^{-1}$, respectively). Modelling results localize the source of high-$T$ (approx. 800°C) gas emissions at a maximum subvolcanic depth of approximately 600 m that further decreases to 200 m for low-$T$ (approx. 100°C) gas emissions (figure 5).

Importantly, the $^{220}Rn/^{222}Rn$ ratio measured in natural settings monotonically decreases with increasing $CO_2$ flux (figure 5), thus corroborating the role played by strong dilution effects on the radioactive signal. There are volcanic conditions in which the rate of $CO_2$ flowing through permeable (fractured/vesiculated) lithologies is high enough to overwhelm the radiogenic source [2,3]. However, some different evolutionary trends are also observed, as long as the $^{220}Rn/^{222}Rn$ ratio increases owing to an increasing permeability/fracturing of the shallow soil/rock (figure 5). Local changes in the soil/rock structure can be interrelated to the presence of preferential pathways for advective $^{220}Rn$ transport via carrier gases [47]. Increasing rates of gas discharge at very shallow depths are viable mechanisms to explain abrupt changes in $^{220}Rn$ emissions, especially along fault zones where $CO_2$ advection is expected to bring $^{220}Rn$ atoms to the ground over a short time scale, thus enhancing the recorded radioactive signal [2,3].

Such different degassing behaviour confirms that chemical and physical properties of the substrate may change significantly as a function of the local geology, thus making unequivocal interpretation of the radioactive signal difficult. As discussed earlier, the changes of $^{220}Rn$ and $^{222}Rn$ concentrations in volcanic environments account for complex mechanisms that cannot be accurately isolated and ascribed to the magnitude of one specific time- and spatial-dependent geochemical variation. In the light of this consideration, figure 5 represents a schematic and conservative overview of more complex geological and radioactive processes. Bearing in mind the lithological and local complexities of volcano-tectonic settings, the experimental $CO_2$ flux threshold from our experiments does not exhibit a universal relationship valid for the interpretation of geochemical anomalies in natural environments. We conclude that the primary requirement for appreciating radon (*sensu lato*) dilution phenomena is the presence of highly permeable lithologies, such as the lithophysae-rich TRSN (approx. 47% porosity) or moderately to extensively fractured crystalline rocks, in conjunction with a high advective gas transport related to abundant degassing from deep-seated magmatic sources. In a holistic perspective, a remarkable decrease of radioactive signal becomes observable when the detrimental-dilution effects of $CO_2$ fluxes greatly exceed any other enhancing agent for the radioactive source.

## 4. Conclusion

Experiments from this study document that the enhancing effect of $^{220}Rn$ signal caused by thermally activated devolatilization reactions in a zeolitized tuff is maximized by the carrier effect of relative low $CO_2$ fluxes. This condition facilitates the expulsion of water molecules and hydroxyl groups adsorbed on the glassy surface of macro- and micropores. Conversely, at higher $CO_2$ flux rates, the increase of $^{220}Rn$ by advective $CO_2$ transport is counterbalanced by strong dilution phenomena of radionuclides within the carrier gas. Zeolite dehydration and $CO_2$ flux represent perturbation mechanisms leading to contrasting non-equilibrium conditions for the activity concentration of $^{220}Rn$. In particular, $CO_2$ degassing may increase or decrease the $^{220}Rn$ radioactive signal, whose sign and magnitude depend on the $CO_2$ flux rate. Because $^{220}Rn$ emissions through a succession of different subvolcanic rocks are mediated over the contributions of magmatic and crustal processes, the $CO_2$ flux threshold responsible for radioactive dilution cannot be univocally quantified by laboratory measurements. The association between highly permeable lithologies and large-scale diffusive $CO_2$ degassing in volcanic areas appears as a viable mechanism for the radon (*sensu lato*) decrease, by overprinting any other enhancing effect related to the radioactive source [2,3]. However, principal component analysis of an entire multi-parametric dataset from volcanic areas (e.g. deep and biogenic $CO_2$ fluxes, $CO_2$ concentration, temperature, $^{222}Rn/^{220}Rn$ ratio) is recommended to reduce dimensionality in the dataset by selecting those variables that mainly control the variance in the geochemical signal and thus discriminating between a deep and a shallow degassing component [6].

Data accessibility. The experimental data of this article have been uploaded as the electronic supplementary material.
Authors' contributions. S.M., P.M. and P.T. designed and performed the radon experiments. G.G. performed the experiments and applied correction to the radon signal. G.I. performed the mineralogical and petrographical analyses. C.L. and P.S. contributed to writing the manuscript. All the authors give final approval for publication.
Competing interests. The authors declare no competing interests.
Funding. No research grant or funding agency supported this research.
Acknowledgements. The authors would like to thank M. Nazzari for the help with laboratory activities and B.S. Ellis for the careful and extensive revision of the English text. Two anonymous reviewers and the Associate Editor, R. Avanzinelli, are acknowledged for their insightful comments and valuable suggestions. The authors are also grateful to G. Bini for a useful discussion about $CO_2$ fluxes in volcanic areas.

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
