## [Peer Review File · Royal Society Open Science]

Review History

RSOS-201539.R0 (Original submission)

Review form: Reviewer 1

Is the manuscript scientifically sound in its present form?

No

Are the interpretations and conclusions justified by the results?

No

Is the language acceptable?

Yes

Do you have any ethical concerns with this paper?

No

Have you any concerns about statistical analyses in this paper?

No

Recommendation?

Major revision is needed (please make suggestions in comments)

Comments to the Author(s)

The paper is interesting but it could be accepted only with major revisions.

The experimental part of the manuscript is very good. The authors however should specify if the pump of the RAD 7 is off or not during experiments and also they should indicate the diameter and height of the rock specimen.

The paragraph 3.3 Limitations and implications for natural volcanic settings and the Conclusion are unacceptable.

In fact all the considerations about Rn-220 do not take in account that the half life of this radionuclide is < 1 minute.

The source of Rn-220 also in a volcanic context cannot be deeper than few meters. In fact the velocity of gas rarely exceed 100 meters per day (see for example : Neri, M., E. Ferrara, S. Giammanco, G. Currenti, R. Cirrincione, G. Patanè, and V. Zanon (2016), Soil radon measurements as a potential tracer of tectonic and volcanic activity, *Sci. Rep.*, 6, 24581, doi:10.1038/srep24581.) Also supposing a velocity ten times higher (1000 meters per day), in five minutes (the time required for the decay of almost all the thoron) the gas travels not more than 4 meters. In other words thoron is useful as tracer of very shallow phenomena. Also the conversion of the experimental data to the field is incorrect. Their flux corresponds to tents of thousands of grams for square meter per day (and not less than 100 as it appears in the figure 5) since their specimens have a diameter(from the paper of Mollo et al.2012, reference 47) of 0.060 meters.

However the dilution effect could be interesting if referred to Rn-222. So I encourage the authors to rewrite the paragraph and conclusion at the light of these considerations.

Review form: Reviewer 2

Is the manuscript scientifically sound in its present form?

No

Are the interpretations and conclusions justified by the results?

No

Is the language acceptable?

No

Do you have any ethical concerns with this paper?

No

Have you any concerns about statistical analyses in this paper?

No

Recommendation?

Major revision is needed (please make suggestions in comments)

Comments to the Author(s)

The results obtained in the present study are innovative and merits publications after due revisions, as suggested (see Appendix A).

Decision letter (RSOS-201539.R0)

Dear Dr Mollo,

The Editors assigned to your paper RSOS-201539 "Carrier and dilution effects of CO₂ on thoron emissions from a zeolitised tuff exposed to sub-volcanic temperatures" have now received comments from reviewers and would like you to revise the paper in accordance with the reviewer comments and any comments from the Editors. Please note this decision does not guarantee eventual acceptance.

Note that the comments of Reviewer 2 were contained in an attached file to which you should have access.

Please submit your revised manuscript and required files (see below) no later than 21 days from today's (ie 08-Dec-2020) date. Note: the ScholarOne system will 'lock' if submission of the revision is attempted 21 or more days after the deadline. If you do not think you will be able to meet this deadline please contact the editorial office immediately.

on behalf of Dr Riccardo Avanzinelli (Associate Editor) and Peter Haynes (Subject Editor)
 openscience@royalsociety.org

Associate Editor Comments to Author (Dr Riccardo Avanzinelli):

Dear authors, your manuscript was carefully evaluated by two reviewers. Both reviewers recognised the quality of the experimental work, its innovative nature and the overall interest for the scientific community. However, both reviewer highlighted major shortcomings and several aspect of the manuscripts that need to be extensively modified, hence both suggesting that major revisions are needed before publications.

From my own reading of the paper I agree with the reviewer comments, in particularly regarding the extremely short half-life of ^{220}Rn , how it may affect the interpretation of the natural data and how (and if) it has been considered when discussing the cases in section 3.3.

As a personal comment, I think it would be really interesting to compare (especially on the natural cases) the behaviour of ^{220}Rn with that of ^{222}Rn (which has a longer half-life) to discuss the effect of decay.

Given the innovative aspects and the quality of the experimental work, I agree with the reviewers that the manuscript will be suitable for publication on Royal Society Open Science after major revisions.

Reviewer comments to Author:

Reviewer: 1

Comments to the Author(s)

The paper is interesting but it could be accepted only with major revisions.

The experimental part of the manuscript is very good. The authors however should specify if the pump of the RAD 7 is off or not during experiments and also they should indicate the diameter and height of the rock specimen.

The paragraph 3.3 Limitations and implications for natural volcanic settings and the Conclusion are unacceptable.

In fact all the considerations about Rn-220 do not take in account that the half life of this radionuclide is < 1 minute.

The source of Rn-220 also in a volcanic context cannot be deeper than few meters. In fact the velocity of gas rarely exceed 100 meters per day (see for example : Neri, M., E. Ferrera, S. Giammanco, G. Currenti, R. Cirrincione, G. Patanè, and V. Zanon (2016), Soil radon measurements as a potential tracer of tectonic and volcanic activity, *Sci. Rep.*, 6, 24581, doi:10.1038/srep24581.) Also supposing a velocity ten times higher (1000 meters per day), in five minutes (the time required for the decay of almost all the thoron) the gas travels not more than 4 meters. In other words thoron is useful as tracer of very shallow phenomena. Also the conversion of the experimental data to the field is incorrect. Their flux corresponds to tents of thousands of grams for square meter per day (and not less than 100 as it appears in the figure 5) since their specimens have a diameter(from the paper of Mollo et al.2012, reference 47) of 0.060 meters.

However the dilution effect could be interesting if referred to Rn-222 . So I encourage the authors to rewrite the paragraph and conclusion at the light of these considerations.

Reviewer: 2
Comments to the Author(s)

Please see attached file. The results obtained in the present study are innovative and merits publications after due revisions, as suggested.

===PREPARING YOUR MANUSCRIPT===

===PREPARING YOUR REVISION IN SCHOLARONE===

<https://royalsociety.org/journals/authors/author-guidelines/#supplementary-material> to include a suitable title and informative caption. An example of appropriate titling and captioning may be found at https://figshare.com/articles/Table_S2_from_Is_there_a_trade-off_between_peak_performance_and_performance_breadth_across_temperatures_for_aerobic_sc_ope_in_teleost_fishes_/3843624.

Author's Response to Decision Letter for (RSOS-201539.R0)

See Appendix B.

Decision letter (RSOS-201539.R1)

The editorial office reopened on 4 January 2021. We are working hard to catch up after the festive break. If you need advice or an extension to a deadline, please do not hesitate to let us know -- we

will continue to be as flexible as possible to accommodate the changing COVID situation. We wish you a happy New Year, and hope 2021 proves to be a better year for everyone.

Dear Dr Mollo,

It is a pleasure to accept your manuscript entitled "Carrier and dilution effects of CO₂ on thoron emissions from a zeolitised tuff exposed to subvolcanic temperatures" in its current form for publication in Royal Society Open Science. The comments of the reviewer(s) who reviewed your manuscript are included at the foot of this letter.

Kind regards,

Anita Kristiansen
Editorial Coordinator

on behalf of Dr Riccardo Avanzinelli (Associate Editor) and Peter Haynes (Subject Editor)
openscience@royalsociety.org

Associate Editor Comments to Author (Dr Riccardo Avanzinelli):
Comments to the Author:
Dear Dr Mollo,

I carefully read through the revised version of the manuscript and your rebuttal letter. I found the paper significantly improved and I believe that the changes made to the manuscript and figures successfully address all the reviewers' comments and criticism. Therefore the manuscript in my opinion is now suitable for publication on Royal Society Open Science.

Appendix A

Title of the Manuscript: “Carrier and dilution effects -----volcanic temperatures” by Silvio Mollo et. al.

Submitted to Royal Society Open Science

Reviewer’s Comments:

Overall /General Comments:

The present Manuscript requires considerable compression and reduction in all the relevant aspects in the Section on **Introduction**, in the text part on **Methods (Experimental systems and conditions)** and **Results** and Discussion. This is primarily because quite a few aspects are repeated and also well known based on earlier reports and relevant publications.

Individual Comments:

1. The Introduction deals with Radon studies in a region with zeolitized tuff and variable carbon-dioxide fluxes. It has been shown by a large number of research studies where Carbon-dioxide concentration is responsible for transport of radon as carrier gas, see the publication by G. Etiope and S. Lombardi, Measurement of radon transported by carrier gas through faulted clays in Italy. Jour, of Radioanalytical and Nucl. Chemistry, 193, 1995. There are a large number of other publications, as well as, review papers, where the transport phenomena by carrier gases for Radon/Thoron gas (fluxes) has been discussed. The conjecture/proposition, that variable or low concentration of carbon dioxide can increase or decrease ,the radon fluxes due to degassing, needs to be established. Are there any other evidences to supplement the same, specially in terms of the in-situ field data, in the area of study. Are there any boreholes in proximity, where the carrier gases can be monitored (temporal variability) and the results provided to supplement the same? An enhanced carbon-dioxide or other

carrier gases can result in elevated levels of Radon. Since the present work is primarily on Thoron, an isotope of Radon, with reasonably low half-life. a rapid in-situ measurement in typical study areas may be meaningful if the Radon and Thoron (gas) fluxes are normalised to the other/typical carrier gas concentration in similar lithology, along with other proxy data, at well defined intervals (grid spacing/suitable trend lines) across the fault zones. This would help to elucidate the results i.e. any decrease in thoron concentration during degassing, with adequate reliability. Could the thoron fluxes be attributed to presence of thorium minerals, at shallow depths?

Methods

1. The text part needs to be compressed a lot. as quite a lot of aspects, as mentioned there-in, are already well known to the readers. Only essential aspects as relevant for the present study, should be provided. Before the Section on Methods, a relevant aspect on the field studies viz. Lithology with structural aspects and the faults, fractures and lineaments (if any) in the study area, should have been briefly discussed, including a stratigraphic section. This would add to the significance of the present study and the relevant implications. I feel that this is very much needed.
2. A multi-parametric approach based on other gases viz. sulphur dioxide and carbon isotopes, coupled to Remote Sensing data to delineate the nature of fracture zones and Non-invasive geophysical studies specially for permeability of the tuffs in close proximity to the fracture zone, as possible. This would elucidate between the shallow and deep degassing phenomena and the possible pathways and role of carbon dioxide for transport of radon and specially thoron.

Results & Discussion

1. The first two sentences, what are the typical minerals in the zeolitized tuff? Are some of the glassy (amorphous) in nature. Does the degassing phenomena of the precursor gases depend only on porosity or permeability could also play an effective role?
2. Section 3.2 and 3.3 respectively needs to be compressed and presented/discussed in few relevant paragraphs.

Conclusion

1. The conclusion obtained from the present work is quite unique, except for few earlier preliminary studies. Hence the nature of the 'gas degassing' and the role of the macro and micro-pores on the surface (matrix) needs to be substantiated by allied studies which are recently being used quite extensively.
2. Suitable statistical analysis on the data if undertaken and provided, would be instructive

Appendix B

Dear Dr Mollo,

Please submit your revised manuscript and required files (see below) no later than 21 days from today's (ie 08-Dec-2020) date. Note: the ScholarOne system will 'lock' if submission of the revision is attempted 21 or more days after the deadline. If you do not think you will be able to meet this deadline please contact the editorial office immediately.

Best regards,

Lianne Parkhouse

Editorial Coordinator

Associate Editor Comments to Author (Dr Riccardo Avanzinelli):

Dear authors, your manuscript was carefully evaluated by two reviewers. Both reviewers recognised the quality of the experimental work, its innovative nature and the overall interest for the scientific community. However, both reviewer highlighted major shortcomings and several aspect of the manuscripts that need to be extensively modified, hence both suggesting that major revisions are needed before publications.

From my own reading of the paper I agree with the reviewer comments, in particularly regarding the extremely short half-life of ^{220}Rn , how it may affect the interpretation of the natural data and how (and if) it has been considered when discussing the cases in section 3.3.

As a personal comment, I think it would be really interesting to compare (especially on the natural cases) the behavior of ^{220}Rn with that of ^{222}Rn (which has a longer half-life) to discuss the effect of decay.

Given the innovative aspects and the quality of the experimental work, I agree with the reviewers that the manuscript will be suitable for publication on Royal Society Open Science after major revisions.

Dear Editor,

We are very grateful to both the Reviewers for their valuable work and the efforts done in order to improve the quality of our manuscript. We found the Reviewers' comments very insightful and constructive. We closely agree with all of them and the manuscript has been revised accordingly. We have also taken into strong consideration your thoughtful suggestions and, for this reason, you will find the revised text colored in violet (Associate Editor), blue (Reviewer#1), and red (Reviewer#2). We believe that the manuscript has been substantially improved and can be now considered for publication in Royal Society Open Science.

Sincerely,

Silvio Mollo and co-authors

Reviewer#1

The paper is interesting but it could be accepted only with major revisions. The experimental part of the manuscript is very good.

We are pleased to know that the Reviewer#1 has appreciated our work. Many thanks for this kind comment.

The authors however should specify if the pump of the RAD 7 is off or not during experiments and also they should indicate the diameter and height of the rock specimen.

We have now specified that the pump of RAD 7 was on during the experiments and that the dimensions of the samples were 60 mm in diameter and 200 mm in length for total weight of 1,784 g.

The considerations about Rn-220 do not take in account that the half life of this radionuclide is < 1 minute. The source of Rn-220 also in a volcanic context cannot be deeper than few meters. In fact the velocity of gas rarely exceed 100 meters per day (see for example : Neri, M., E. Ferrera, S. Giammanco, G. Currenti, R. Cirrincione, G. Patanè, and V. Zanon (2016), Soil radon measurements as a potential tracer of tectonic and volcanic activity, Sci. Rep., 6, 24581, doi:10.1038/srep24581.). Also supposing a velocity ten times higher (1000 meters per day), in five minutes (the time required for the decay of almost all the thoron) the gas travels not more than 4 meters. In other words thoron is useful as tracer of very shallow phenomena.

We closely agree with this very important consideration. The manuscript has been now revised on the basis of these parameters, by stressing the very short half-life of Rn-220 in volcanic settings.

Their flux corresponds to tents of thousands of grams for square meter per day (and not less than 100 as it appears in the figure 5) since their specimens have a diameter (from the paper of Mollo et al.2012, reference 47) of 0.060 meters.

Reviwer#1 is totally correct. This notion has been now reported in the revised manuscript.

However the dilution effect could be interesting if referred to Rn-222. So I encourage the authors to rewrite the paragraph and conclusion at the light of these considerations.

Thanks!

- - - -

Reviewer#2

The present Manuscript requires considerable compression and reduction in all the relevant aspects in the Section on Introduction, in the text part on Methods (Experimental systems and conditions) and Results and Discussion. This is primarily because quite a few aspects are repeated and also well known based on earlier reports and relevant publications.

We closely agree with the Reviewer#2's comment. The manuscript has been shortened.

Individual Comments:

The Introduction deals with Radon studies in a region with zeolitized tuff and variable carbon-dioxide fluxes. It has been shown by a large number of research studies where Carbon dioxide concentration is responsible for transport of radon as carrier gas, see the publication by G. Etiope and S. Lombardi, Measurement of radon transported by carrier gas through faulted clays in Italy. Jour, of Radioanalytical and Nucl. Chemistry, 193, 1995.

We thanks the Reviewer#2 for suggesting the study by Etiope and Lombardi. This work and its main conclusions have been now added to the revised text.

There are a large number of other publications, as well as, review papers, where the transport phenomena by carrier gases for Radon/Thoron gas (fluxes) has been discussed. The conjecture/proposition, that variable or low concentration of carbon dioxide can increase or decrease ,the radon fluxes due to degassing, needs to be established. Are there any other evidences to supplement the same, specially in terms of the in-situ field data, in the area of study. Are there any boreholes in proximity, where the carrier gases can be monitored (temporal variability) and the results provided to supplement the same?

To our knowledge, there are no Rn-CO2 monitored boreholes in proximity of the Vico volcanic apparatus (Latium, Italy) from which the TRSN tuff has been sampled. However, in terms of in-situ

field data, there are two studies on Mt. Etna volcano where the increasing/decreasing activity concentrations of Radon/Thoron have been related to variable CO₂ flux from the source rock. We have now improved the Introduction Section with this information.

An enhanced carbon-dioxide or other carrier gases can result in elevated levels of Radon. Since the present work is primarily on Thoron, an isotope of Radon, with reasonably low half-life, a rapid in-situ measurement in typical study areas may be meaningful if the Radon and Thoron (gas) fluxes are normalised to the other/typical carrier gas concentration in similar lithology, along with other proxy data, at well defined intervals (grid spacing/suitable trend lines) across the fault zones. This would help to elucidate the results i.e. any decrease in thoron concentration during degassing, with adequate reliability.

Many thanks for the suggestion. The different behavior of Radon and Thoron as a function their half-lives has been now discussed in Section 3.3, also following the comments of the Editor and Reviewer#1.

Could the thoron fluxes be attributed to presence of thorium minerals, at shallow depths?

We do not exclude this possibility and find it highly reliable. Section 3.3 has been revised accordingly. Thanks again for this valuable suggestion.

Methods

The text part needs to be compressed a lot. as quite a lot of aspects, as mentioned there-in, are already well known to the readers. Only essential aspects as relevant for the present study, should be provided.

We have shortened the text by avoiding eventual repetitions.

Before the Section on Methods, a relevant aspect on the field studies viz. Lithology with structural aspects and the faults, fractures and lineaments (if any) in the study area, should have been briefly discussed, including a stratigraphic section. This would add to the significance of the present study and the relevant implications.

Since 2008 we investigate in laboratory how the chemical and physical changes of rocks may influence the radon emissions. The rock types are selected on the basis of their petrophysical characteristics and not whether they belongs to an area where geochemical and geophysical parameters are monitoring. For the case of the TRSN tuff from Vico volcanic apparatus, we do not have knowledge of specific areas where radon emissions from this tuff are measured in the field. Therefore, the aim of our work is to better understand the empirical relationship between thorn activity concentration and CO₂ flux. We have now emphasized this aspect in the introduction section of the manuscript. Moreover, in our final conceptual model we have explained how structural aspects and faults in an hypothetical natural system may favor CO₂ degassing and radon/thoron emissions.

A multi-parametric approach based on other gases viz. sulphur dioxide and carbon isotopes, coupled to Remote Sensing data to delineate the nature of fracture zones and Non-invasive geophysical studies specially for permeability of the tuffs in close proximity to the fracture zone, as possible. This would elucidate between the shallow and deep degassing phenomena and the possible pathways and role of carbon dioxide for transport of radon and specially thoron.

Many thanks for this thoughtful suggestion. As reported above, there are no monitoring data that we can use for the area where the TRSN tuff outcrops, as well as the elaboration of geochemical data from multi-parametric geochemical surveys is beyond the main scope of our experimental work conducted in laboratory. However, we found the suggestion of Reviewer#2 very useful to improve the Discussion Section 3.3 and to stress the importance of a multi-parametric approach to better decipher the shallow and deep degassing phenomena in an hypothetical studied area.

Results & Discussion

The first two sentences, what are the typical minerals in the zeolitized tuff? Are some of the glassy (amorphous) in nature.

We have now improved the petrographic description of the tuff rock.

Does the degassing phenomena of the precursor gases depend only on porosity or permeability could also play an effective role?

We have now discussed in the text the control of rock porosity/permeability on radon emissions.

Section 3.2 and 3.3 respectively needs to be compressed and presented/discussed in few relevant paragraphs.

This has been now done. Thanks for the valuable suggestion.

Conclusion

The conclusion obtained from the present work is quite unique, except for few earlier preliminary studies. Hence the nature of the 'gas degassing' and the role of the macro and micro-pores on the surface (matrix) needs to be substantiated by allied studies which are recently being used quite extensively.

This has been now done.

Suitable statistical analysis on the data if undertaken and provided, would be instructive.

We did not collect natural geochemical data and, therefore, a statistical analysis cannot be performed on them. Our work is mainly focused on empirical parameters measured in laboratory. However, we have now added to the Conclusion Section a further remark in which Principal Component Analysis (PCA) of an entire multi-parametric data set from volcanic areas (e.g., deep and biogenic CO₂ fluxes, CO₂ concentration, temperature, ²²²Rn/²²⁰Rn ratio) is recommended to dimensionality in the data set by selecting those variables that mainly control the variance in the geochemical signal, thus between a deep and a shallow degassing component.